# S4G: Breaking the Bottleneck on Graphs with Structured State Spaces

## Abstract

The majority of GNNs are based on message-passing mechanisms, however, message-passing neural networks (MPNN) have inherent limitations in capturing long-range interactions. The exponentially growing node information is compressed into fixed-size representations through multiple rounds of message passing, bringing the over-squashing problem, which severely hinders the flow of information on the graph and creates a bottleneck in graph learning. The natural idea of introducing global attention to point-to-point communication, as adopted in graph Transformers (GT), lacks inductive biases on graph structures and relies on complex positional encodings to enhance their performance in practical tasks. In this paper, we observe that the sensitivity between nodes in MPNN decreases exponentially with the shortest path distance. Contrarily, GT has a constant sensitivity, which leads to its loss of inductive bias. To address these issues, we introduce structured state spaces to capture the hierarchical structure of rooted-trees, achieving linear sensitivity with theoretical guarantees. We further propose a novel graph convolution based on the state-space model, resulting in a new paradigm that retains both the strong inductive biases from MPNN and the long-range modeling capabilities from GT. Extensive experimental results on long-range and general graph benchmarks demonstrate the superiority of our approach.[1]

## 1 Introduction

Graph neural networks (GNNs) (Scarselli et al., 2008; Bruna et al., 2013; Defferrard et al., 2016; Kipf & Welling, 2017; Hamilton et al., 2017; Gilmer et al., 2017; Veličković et al., 2018; Xu et al., 2019) have rapidly developed in recent years and are now the dominant method for learning representations for graphs. Most GNNs rely on the message-passing mechanism (Gilmer et al., 2017) to encode both node features and the graph structure. However, it has been shown that message-passing neural networks (MPNNs) struggle with modeling the long-range interactions (LRI) on graphs (Alon & Yahav, 2021; Topping et al., 2022; Dwivedi et al., 2022b). MPNNs only exchange information with neighboring nodes, requiring multiple stacked layers to interact with distant nodes. Unfortunately, this leads to the *vanishing gradients* and hinders proper model training (Li et al., 2019; Zhao & Akoglu, 2020). Moreover, stacking multiple layers of MPNNs causes node representations to converge excessively, resulting in the *over-smoothing* problem (Li et al., 2018) that negatively impacts the model's generalization performance. Alon & Yahav (2021) identified the main culprit hindering MPNNs from effectively handling LRI as the *over-squashing* problem: as more layers are stacked in MPNN, information from exponentially more nodes is compressed into a fixed-size node representation. Topping et al. (2022) indicates that in multi-layer MPNNs, the sensitivity between nodes decreases exponentially with the shortest path distance, which fundamentally limits the ability of MPNN and creates a *bottleneck* for learning on graphs.

In order to break the bottleneck on graphs and improve the learning of GNNs, a large number of methods have been proposed, including multi-hop MPNN (Xu et al., 2018; Abu-El-Haija et al., 2019; Abboud et al., 2022), graph Transformer (GT) (Dwivedi & Bresson, 2020; Kreuzer et al., 2021; Ying et al., 2021), and graph rewiring (Klicpera et al., 2019; Topping et al., 2022). *Multi-hop MPNN* carefully integrates information from multiple hops of neighbors (Xu et al., 2018) or allows nodes to directly access higher-order neighbors at each layer (Abu-El-Haija et al., 2019).

---

[1] Code will be released at `anonymous.url.com`

Although these methods can effectively alleviate the over-smoothing problem and improve gradient propagation since they provide various shortcuts, they often rely on the powers of normalized adjacency matrices, which is exactly what causes the over-squashing problem. Abboud et al. (2022) directly aggregate messages from shortest path and breaking the bottleneck, despite partially lose the structural information. *Graph Transformer* achieves strong general modeling capabilities by introducing a global attention mechanism (Vaswani et al., 2017) to free the model from the constraints of the graph structure, allowing each node in the graph to directly access all other nodes, albeit at the expense of incorporating less inductive biases from the graph structure. Thus it shows poor performance on many graph tasks (Ying et al., 2021). Researchers have attempted to inject structural information through position encoding (PE) (Dwivedi & Bresson, 2020; Kreuzer et al., 2021) or structural encoding (SE) (Rampášek et al., 2022). But unlike regular-shaped data such as sequences or images, designing effective PE for non-Euclidean graph data is very challenging and often relies on heuristics. Different from the previous two types of methods, *graph rewiring* focuses on the data aspect by attempting to modify the graph structure for better message passing.[2] These methods involve deleting or adding edges on a graph based on a predefined optimization objective. They rely heavily on prior knowledge and are unable to learn the modification strategy.

In this paper, we start with the node-pair sensitivity analysis and point out the difference between MPNN and GT. In MPNN, the sensitivity decreases *exponentially* with distance, while in GT, the sensitivity between nodes does not change with distance, i.e., GT has a *constant* sensitivity. We believe that this is a key reason for GT's loss of inductive bias on graphs. Based on this observation, we aim to design a model that *strike a balance between these two extremes of sensitivity*. Structured state spaces (S4) (Gu et al., 2022) provide such a principled way. S4 is a variant of state-space models (SSM) (Gu et al., 2021), and makes breakthroughs in learning long-range dependencies on sequences. They provide many good properties, such as the ability to memorize the entire history of the sequence, which could be useful for the over-smoothing problem since the main cause of it is the loss of early and local information. More importantly, S4 provides stable gradients that decay linearly with sequence length, which helps to overcome the RNN bottleneck. This could improve the gradients between nodes for graph learning, improving vanishing gradients and over-squashing.

By introducing structured state spaces for graphs, we achieve a *linear* decrease in node-pair sensitivity. We then propose S4G-Conv, a graph convolution based on SSM. It makes applying SSM on graphs computationally feasible and allows each node to directly access its high-order neighbors. By integrating the S4G-Conv into a Transformer-like architecture (without attention), we propose the S4G model. Our model exhibits linearly decaying sensitivity instead of exponentially, transmitting information on graphs more smoothly and breaking the bottleneck on graphs. Additionally, our model preserves strong inductive bias and performs well on graph tasks without requiring any positional encoding or structural encoding.

## 2 SENSITIVITY ANALYSIS

In this section, we use sensitivity analysis to investigate the issue of over-squashing in MPNNs. We also explore the sensitivity of GTs and the relationship to the loss of inductive bias on graphs. Based on this analysis, we introduce the insights that motivate our model design.

Let's consider an unweighted, undirected, and connected graph $\mathcal{G}$, where $\mathcal{V}$ represents the nodes and $\mathcal{E}$ represents the edges, $(i, j) \in \mathcal{E}$ when node $i$ and $j$ are direct neighbors on $\mathcal{G}$. The adjacency matrix of $\mathcal{G}$ is denoted as $A$. $\tilde{A} = A + I$ and $\tilde{D} = D + I$ denote augmenting the adjacency matrix $A$ and degree matrix $D$ with self-loops, respectively. By applying the normalization $\hat{A} = \tilde{D}^{-\frac{1}{2}} \tilde{A} \tilde{D}^{-\frac{1}{2}}$, we obtain the widely used normalized augmented adjacency matrix $\hat{A}$. We denote the shortest path distance between nodes $i, j$ as $d_{\mathcal{G}}(i, j)$, the $k$-hop neighborhood for node $i$ is $\mathcal{N}_k(i) = \{j \mid d_{\mathcal{G}}(i, j) = k\}$, and neighborhood within $k$-hops as $\mathcal{B}_k(i) = \{j \mid d_{\mathcal{G}}(i, j) \leq k\}$. For simplicity, we assume the node feature $h_i$ to be a scalar following Topping et al. (2022); Gu et al. (2020), analogous results can be obtained in vector case. We define the *sensitivity* between nodes $i, j$ as the norm of the Jacobian $\|\partial h_i / \partial h_j\|$ when they interact with each other *for the first time*. Note that the $h_i, h_j$ can be in different model layers, we will add superscripts to them when it happens.

---

[2]Gutteridge et al. (2023) encompasses multi-hop MPNN and GT into the scope of graph rewiring with a broader definition. We discuss graph rewiring in a narrower sense here to distinguish it from other methods.

**MPNNs and their sensitivity.** Typically, the message-passing process can be partitioned into two stages: the *aggregate* stage and the *combine* stage (Gilmer et al., 2017). In the $\ell$-th layer, given node $i$, it first aggregates information from its immediate neighbors $\mathcal{N}_1(i)$, generating the message $m_i^{(\ell)}$. Subsequently, $m_v^{(\ell)}$ is combined with the ego representation $h_i^{(\ell-1)}$ from the previous layer. Formally,

$$h_i^{(\ell)} = \phi_\ell\left(h_i^{(\ell-1)}, m_i^{(\ell)}\right), m_i^{(\ell)} = \sum_{j=1}^{|\mathcal{N}_1(i)|} \hat{A}_{ij}\psi_\ell\left(h_i^{(\ell-1)}, h_j^{(\ell-1)}\right) \tag{1}$$

The first equation represents the *combine* stage, while the second one for the *aggregate* stage. For nodes $i, j$ with distance $d_\mathcal{G}(i, j) = r$, we need to stack $r$ layers to allow them to *start* interacting. (Topping et al., 2022) proves that with bounded derivatives for functions in Equation 1: $|\nabla\phi_\ell| \leq \alpha$ and $|\nabla\psi_\ell| \leq \beta$, we have sensitivity $\left\|\partial h_i^{(r)}/\partial h_j^{(0)}\right\| \leq (\alpha\beta)^r(\hat{A}^r)_{ij}$.[3] We can see for a given distance $r$, it depends on two terms simultaneously: the property of the functions in MPNN and the property of the graph structure. If $\alpha\beta < 1$, the model would experience the classical issue of gradient vanishing, which is also observed in RNNs, is caused by the model itself. However, what's even more concerning is that the second term causes sensitivity to *exponential decay once again* due to graph topology. This makes the bottleneck on graphs much more hazardous.

**GTs and their sensitivity.** In this study, we focus on the vanilla Transformer architecture (Vaswani et al., 2017), equipped with graph PE. The vanilla Transformer consists of two blocks: multi-head attention (MHA) and feed-forward network (FFN). Due to the fact that each node in MHA can interact with all other nodes, and FFN operates *independently* on each node, for simplicity purposes, we only analyze the sensitivity of key-query-value (KQV) attention, which is the core of MHA. We present the KQV attention with graph PE here:

$$h_i^{'} = \sum_{j=1}^{|\mathcal{V}|} a_{ji}f_V\left(h_j + p_j\right), a_{ji} = \frac{\exp\left(f_K\left(h_j + p_j\right)f_Q\left(h_i + p_i\right)\right)}{\sum_{k=1}^{|\mathcal{V}|}\exp\left(f_K\left(h_k + p_k\right)f_Q\left(h_i + p_i\right)\right)} \tag{2}$$

Here, $h_i^{'}$ represents the output embedding for node $i$, while $f_K, f_Q$, and $f_V$ refer to the key, query, and value mapping functions respectively. In GT literature, graph PE $p_i$ is usually injected into node embeddings in an absolute PE fashion (Dwivedi et al., 2022a), thus we add them to $h_i$ here. Common graph PE include Random Walk PE (Dwivedi et al., 2022a), Laplacian PE (Dwivedi et al., 2020), etc. They are computed based on the graph topology, thus $p_i$ is the function of $A$ and $i$: $p_i = p(A, i)$. Considering $p_i = (D^{-1}A)_{ii}^K$ the Random Walk PE, although the value of it decays exponentially with random walk steps $K$, $p_i = p(A, i)$ is not the function of neighbor node $h_j$ and its derivatives would be 0. Furthermore, GT do not include the consecutive multiplication of the functions, therefore, the sensitivity in GT $\left\|\partial h_i^{'}/\partial h_j\right\| \leq c$. The $c$ here is related to the boundary of three mapping functions and their derivatives. Thus GT has *constant* sensitivity in terms of $r$.

Based on the analysis above, we observe that the node-pair sensitivity of MPNN decreases exponentially with $r$, leading to the over-squashing problem. However, it effectively reflects the graph structure through *gradients between nodes*. On the other hand, GT maintains a constant sensitivity. While this removes the bottleneck, it also eliminates the graph inductive bias. It is worth noting that although injecting the graph PE can enhance GNNs' ability to distinguish isomorphic graphs (Li et al., 2020; Dwivedi et al., 2022a), it does not help recover the distance-based sensitivity. A natural question would be: can we design a model with a sensitivity decay between exponential and constant? Structured state spaces provide such a principled way to achieve a *linear* decay, and we tailor it for graphs.

## 3 STRUCTURED STATE SPACES FOR GRAPHS (S4G)

In this section, we first introduce the structured state spaces, a variant of SSMs with a special state matrix. Then, we show how to extend it to graphs by capturing the inductive bias from rooted trees. Next, we propose an SSM-based convolution on graphs (S4G-Conv) and analyze its theoretical properties. Finally, we propose the full model architecture centered around S4G-Conv.

---

[3]Giovanni et al. (2023) analyzed more general cases, for example, not requiring the adjacency matrix to be normalized.

### 3.1 STRUCTURED STATE SPACES

SSM is a fundamental model in many scientific domains such as control theory. (Gu et al., 2022) proposed a specific instance of it, structured state spaces, achieving breakthroughs in addressing long-range problems for sequential data. Unlike RNNs or Transformers that directly model the sequence, they first model the continuous signal behind it. Given 1-D signal $u(t)$, SSM first maps the signal into a $N$-D latent space $x(t)$ and then projects it into the output 1-D signal $y(t)$.

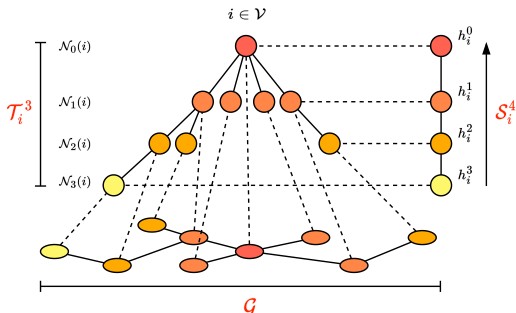

$$x'(t) = \boldsymbol{A}x(t) + \boldsymbol{B}u(t)$$
$$y(t) = \boldsymbol{C}x(t) + \boldsymbol{D}u(t) \qquad (3)$$

$\boldsymbol{A}, \boldsymbol{B}, \boldsymbol{C}, \boldsymbol{D}$ are the parameters of SSM. The basic SSM is simple, the key to addressing the long-range problem is to design the state matrix $\boldsymbol{A}$, thus effectively compressing the $u(t)$ in infinite time into the finite $N$-D state $x(t)$. To achieve this, they use the HiPPO framework (Gu et al., 2020) to initialize $\boldsymbol{A}, \boldsymbol{B}$ with special structure. The basic idea is to treat the contin-

Figure 1: In the rooted-tree hierarchy, for node $i \in \mathcal{V}$, the depth $L$ rooted-tree for it is denoted as $\mathcal{T}_i^L$. We construct the hop-wise sequence $\mathcal{S}_i^{L+1}$ from it through hop-wise aggregation, and append the root node to the end of the sequence. The constructed sequence is then encoded by a structured state space model.

uous signal as a function of $t$, then the signal compression is formulated as an online function approximation problem, which has a long history and mature solutions. Specifically, they propose

$$\boldsymbol{A}_{nk} = - \begin{cases} (2n+1)^{1/2}(2k+1)^{1/2} & \text{if } n > k \\ n+1 & \text{if } n = k \\ 0 & \text{if } n < k \end{cases} \qquad \boldsymbol{B}_n = (2n+1)^{1/2} \qquad (4)$$

It is proved that with these well-designed $\boldsymbol{A}, \boldsymbol{B}$, the state $x(t)$ can *memorize* all the history of $u(t)$, which means the full $u(t), t \in [0, T]$ can be optimally approximated from $N$-D state $x(T)$ with theoretical guarantees, $N$ here can be treated as the order of orthogonal polynomials used for function approximation. Finally, a discretization is applied to the SSM with step size $\Delta$, a scalar parameter for resolution. The discrete structured state spaces for sequential data are presented below. $k$ denote the time step, $\boldsymbol{D}$ in Equation 3 is omitted and can be implemented as a skip-connection.

$$x_k = \overline{\boldsymbol{A}}x_{k-1} + \overline{\boldsymbol{B}}u_k \quad \overline{\boldsymbol{A}} = (\boldsymbol{I} - \Delta/2 \cdot \boldsymbol{A})^{-1}(\boldsymbol{I} + \Delta/2 \cdot \boldsymbol{A})$$
$$y_k = \overline{\boldsymbol{C}}x_k \qquad \overline{\boldsymbol{B}} = (\boldsymbol{I} - \Delta/2 \cdot \boldsymbol{A})^{-1}\Delta\boldsymbol{B} \qquad \overline{\boldsymbol{C}} = \boldsymbol{C} \qquad (5)$$

Due to its strong memory capacity for long sequences, the structured state spaces can effectively handle long-range problems. Another excellent property of the HiPPO framework is its stable gradients. The gradient norm between tokens linearly decays with length instead of exponentially. This solves the problem of vanishing gradients that create the RNN bottleneck. The updating rule for $x_k$ in Equation 5 with $\boldsymbol{A}, \boldsymbol{B}$ defined in Equation 4 is called the HiPPO-LegS operator. We have

**Lemma 1** *(Gu et al., 2020, Proposition 7) For any times $k < \ell$, the gradient norm of the HiPPO-LegS operator for the output at time $\ell + 1$ with respect to input at time $k$ is $\|\partial x_{\ell+1}/\partial u_k\| = \Theta(1/\ell)$.*

### 3.2 CONSTRUCTING HOP-WISE SEQUENCES FROM ROOTED-TREES

Due to the irregular nature of graph data, extending the structured state spaces to graphs is non-trivial. In sequential data, SSM goes through each token in the sentence in order. However, graph data has an unordered structure, making it infeasible to directly apply SSM to graphs. Recent works have shown that node-centric rooted-trees effectively capture structural information on graphs (Zhang & Li, 2021; Chen et al., 2023; Song et al., 2023). The hierarchical structure of rooted-trees naturally involves the order of precedence, making them highly suitable for constructing sequences on graphs.

As shown in Figure 1, from the perspective of a node, the neighborhood within multiple hops naturally forms a rooted-tree. The rooted-tree $\mathcal{T}_i^L$ is a tree with depth $L$ rooted at node $i$. Considering the message-passing mechanism on graphs, it essentially extends the encoded depth of a rooted-tree with multiple rounds. Compared to $k-1$-th round of message passing for node $i$, the $k$-th round would *newly* involve the nodes from $\mathcal{N}_k(i)$, which extend the encoded rooted-tree from $\mathcal{T}_i^{k-1}$ to $\mathcal{T}_i^k$ in a sequential process. We can simulate this process by encoding the increasing depths of rooted-tree sequentially, thus preserving the inductive bias.

We encode a set of nodes $\mathcal{N}_k(i)$ at depth $k$ of $\mathcal{T}_i^L$ using a permutation-invariant function $f$, and get the embedding of $k$-hop neighborhood $h_i^k$, obtaining a hop-wise sequence $\mathcal{S}_i^{L+1}$ of length $L+1$.

$$h_i^k = f\left(\{h_j | j \in \mathcal{N}_k(i)\}\right)$$
$$\mathcal{S}_i^{L+1} = \left(h_i^L, h_i^{L-1}, \cdots, h_i^0\right) \tag{6}$$

Here, we treat the root node embedding $h_i^0 = h_i$ as the last token in the sequence $\mathcal{S}_i^{L+1}$, matching how the information of the entire sequence is compressed into the final token in a sequence model. The selection of function $f$ can be diverse. In this paper, we simply implement it as the add pooling, $f(\cdot) = \text{ADD}(\cdot)$.

### 3.3 S4G-Conv: SSM Convolution on Graphs

After constructing the hop-wise sequences for each node from the rooted-trees, we can use structured state spaces to encode them. Different from the sequential data, each sequence here is only for the central node, thus we only need the representation of the final token, $C$ is only applied to $x_i^L$, while $A, B$ is used in each step. We omit the superscripts indicating the number of layers for simplicity, and $h_i^{'}$ denote the output embedding for node $i$.

$$x_i^0 = \overline{B}h_i^L, x_i^1 = \overline{A}x_i^0 + \overline{B}h_i^{L-1}, \cdots, x_i^L = \overline{A}x_i^{L-1} + \overline{B}h_i^0$$
$$h_i^{'} = \overline{C}x_i^L \tag{7}$$

If we unroll the updating rule for $x_i$, we can obtain a convolution kernel for SSM on graphs. This kernel is independent of the central node and can be computed once and then shared across the graph.

$$h_i^{'} = \overline{CA}^L\overline{B}h_i^L + \overline{CA}^{L-1}\overline{B}h_i^{L-1} + \cdots + \overline{CAB}h_i^1 + \overline{CB}h_i^0$$
$$\overline{K} = (\overline{CB}, \overline{CAB}, \cdots, \overline{CA}^{L-1}\overline{B}, \overline{CA}^L\overline{B}) \tag{8}$$

With this convolution kernel, we can make the SSM-based encoding of rooted-tree feasible, since we can compute the representation of a rooted-tree with once weighted sum, instead of matrix-vector multiplications step by step. Recall we select $\text{ADD}(\cdot)$ as the encoding function for $h_i^k$, combined with Equation 8, we obtain the SSM-based convolution on graphs (S4G-Conv).

$$h_i^{'} = \sum_{k=0}^{L} \sum_{j=1}^{|\mathcal{N}_k(i)|} \overline{K}_{[k]} h_j = \sum_{j=1}^{|\mathcal{B}_L(i)|} \overline{K}_{[d_{\mathcal{G}}(i,j)]} h_j \tag{9}$$

As we can see, S4G-Conv allows each node to directly access its high-order neighbors $j \in \mathcal{B}_L(i)$ within $L$ hops through SSM kernel values. Since the structured state matrix that constructs $\overline{K}$ provides stable gradients, and the functions used for hop-wise encoding do not rely on distance, S4G-Conv also provides such linear decayed gradients, which is exactly the node-pair linear sensitivity.

**Proposition 2** *(Proof in Appendix A) For two nodes $i, j \in \mathcal{G}$ with shortest path distance $d_{\mathcal{G}}(i,j) = r \leq L$, S4G-Conv defined in Equation 9 ensures the sensitivity between $i, j$ to be $\|\partial h_i / \partial h_j\| = \Theta(1/r)$.*

S4G-Conv provides some good properties for learning on graphs. First, with a linear decayed sensitivity in terms of distance, we break the bottleneck that causes *over-squashing*. Second, the *stable gradients* also removes the exponential decay caused by both parameters and graph topology, helping the model learn better. Then, since structured state spaces guarantee a full memory of history, S4G-Conv also captures the per-hop neighborhood information precisely, thus alleviating *over-smoothing*. It is worth noting that compared to global attention with quadratic complexity, we limit the access scope for each node with $L$ hops, this allows us to set a fixed node budget and achieve linear complexity, making it potentially *scalable* when dealing with large graphs.

Table 1: Comparison of S4G with baselines on Tree-Neighbors-Match (Alon & Yahav, 2021).

| Depth | r=2 | r=3 | r=4 | r=5 | r=6 | r=7 | r=8 |
|---|---|---|---|---|---|---|---|
| GGNN (Li et al., 2015) | 1.00 | 1.00 | 1.00 | 0.60 | 0.38 | 0.21 | 0.16 |
| GCN (Kipf & Welling, 2017) | 1.00 | 1.00 | 0.70 | 0.19 | 0.14 | 0.09 | 0.08 |
| GAT (Veličković et al., 2018) | 1.00 | 1.00 | 1.00 | 0.41 | 0.21 | 0.15 | 0.11 |
| GIN (Xu et al., 2019) | 1.00 | 1.00 | 0.77 | 0.29 | 0.20 | - | - |
| **S4G** | **1.00** | **1.00** | **0.99** | **0.98** | **1.00** | **1.00** | **1.00** |

## 3.4 FULL S4G ARCHITECTURE

Since SSMs are linear models that lack nonlinearity, we integrate S4G-Conv into the Transformer architecture for it, we follow the pre-LayerNorm for better training. We directly replace the KQV attention in the MHA block with S4G-Conv, while preserving the rest, and then we get the $\ell$-th S4G layer here. We present the model in the matrix fashion, and $\boldsymbol{H}_{[i]}^{(\ell)} = \boldsymbol{h}_i^{(\ell)}$ is a vector, when applying S4G-Conv to vectors, we simply share the SSM kernel with scalar value to all the dimensions.

$$\tilde{\boldsymbol{H}}^{(\ell)} = \text{Conv}\left(\text{LN}\left(\boldsymbol{H}^{(\ell-1)}\right)\boldsymbol{W}_V^{(\ell)}\right)\boldsymbol{W}_O^{(\ell)} + \boldsymbol{H}^{(\ell-1)}$$
$$\boldsymbol{H}^{(\ell)} = \text{FFN}\left(\text{LN}\left(\tilde{\boldsymbol{H}}^{(\ell)}\right)\right) + \tilde{\boldsymbol{H}}^{(\ell)}$$

(10)

In practice, we find letting $\overline{\boldsymbol{K}}$ to be trainable would cause two problems, one is slow gradient backpropagation, since the gradients from all the nodes need to propagate through $L$-th power of structured state matrices; another one is the trainable state matrix might damage the structure as stated in Equation 4, and break the theoretical guarantee from HiPPO framework (Gu et al., 2022). Thus we apply the *frozen SSM* to our model, let $\overline{\boldsymbol{K}}$ fixed during training, and then the trainable parameters in our model only contain 4 linear layers.

## 4 EXPERIMENTS

In this section, we benchmark our model on 9 real-world datasets and a series of 7 synthetic datasets. We first conduct experiment on a series of synthetic datasets that simulate the bottleneck on graphs, to check if our model could break the bottleneck and overcome over-squashing. Second, we evaluate how our model performs when dealing with LRI on realistic graphs, with 5 real-world datasets designed for LRI. Then, since the proposed S4G doesn't rely on message passing or attention mechanism, we check the generality of this new paradigm on 4 general graph tasks, covering both node-level and graph-level. Finally, we propose an ablation study on S4G-Conv, the core of S4G, and the main difference against Transformer.

**Experimental setup.** For performance evaluation on real-world datasets, we divided the baseline into two parts. One part consists of classical MPNNs and GT (Transformer with graph PE), while the other part includes the SOTA models that incorporate a hybrid architecture of MPNN, GT, and PE/SE. In comparison to the model with a hybrid architecture, in order to validate our model's potential, we further introduce S4G+. Similar to a SOTA model, GraphGPS (Rampášek et al., 2022), we add the output of an MPNN with that of S4G and then perform feature fusion through an MLP. Unlike other hybrid architecture models, we do not use any PE or SE. Additionally, we fix the backbones for MPNN in S4G+, using simple ones. Specifically, we use GCN (Kipf & Welling, 2017)/GatedGCN (Bresson & Laurent, 2017) for datasets without/with edge features, this is because GCN can not process the edge features. While in GraphGPS, they treat MPNN, GT, PE/SE as hyperparameters. For baselines with variants on different datasets, when reporting performance results, we append an asterisk (*) after their names and select the best result from these variants for each dataset. More details can be found in Appendix C.

Table 2: Comparison of S4G, S4G+ with baselines on Long Range Graph Benchmarks (Dwivedi et al., 2022b). Best results are colored in **first**, **second** and **third**.

| Model | PascalVOC-SP F1 score ↑ | COCO-SP F1 score ↑ | Peptides-func AP ↑ | Peptides-struct MAE ↓ | PCQM-Contact MRR ↑ |
|---|---|---|---|---|---|
| GCN | $0.1268 \pm 0.0060$ | $0.0841 \pm 0.0010$ | $0.5930 \pm 0.0023$ | $0.3496 \pm 0.0013$ | $0.3234 \pm 0.0006$ |
| GINE | $0.1265 \pm 0.0076$ | $0.1339 \pm 0.0044$ | $0.5498 \pm 0.0079$ | $0.3547 \pm 0.0045$ | $0.3180 \pm 0.0027$ |
| GatedGCN* | $0.2873 \pm 0.0219$ | $0.2641 \pm 0.0045$ | $0.6069 \pm 0.0035$ | $0.3357 \pm 0.0006$ | $0.3242 \pm 0.0008$ |
| GT | $0.2694 \pm 0.0098$ | $0.2618 \pm 0.0031$ | $0.6326 \pm 0.0126$ | $0.2529 \pm 0.0016$ | $0.3174 \pm 0.0020$ |
| MixHop* | $0.2506 \pm 0.0133$ | - | $0.6843 \pm 0.0049$ | $0.2614 \pm 0.0023$ | $0.3250 \pm 0.0010$ |
| DIGL* | $0.2921 \pm 0.0038$ | - | $0.6830 \pm 0.0026$ | $0.2616 \pm 0.0018$ | $0.1707 \pm 0.0021$ |
| SPN | $0.2056 \pm 0.0338$ | - | $0.6926 \pm 0.0247$ | $0.2554 \pm 0.0035$ | $0.3236 \pm 0.0051$ |
| **S4G** | $0.3183 \pm 0.0078$ | $0.2602 \pm 0.0027$ | $\mathbf{0.7213 \pm 0.0009}$ | $\mathbf{0.2485 \pm 0.0017}$ | $0.3330 \pm 0.0003$ |
| SAN* | $0.3230 \pm 0.0039$ | $0.2592 \pm 0.0158$ | $0.6439 \pm 0.0075$ | $0.2545 \pm 0.0012$ | $0.3350 \pm 0.0003$ |
| GraphGPS | $\mathbf{0.3748 \pm 0.0109}$ | $\mathbf{0.3412 \pm 0.0044}$ | $0.6535 \pm 0.0041$ | $0.2500 \pm 0.0005$ | $0.3337 \pm 0.0006$ |
| Exphormer | $\mathbf{0.3975 \pm 0.0037}$ | $\mathbf{0.3455 \pm 0.0009}$ | $0.6527 \pm 0.0043$ | $\mathbf{0.2481 \pm 0.0007}$ | $\mathbf{0.3637 \pm 0.0020}$ |
| DRew* | $0.3314 \pm 0.0024$ | - | $\mathbf{0.7150 \pm 0.0044}$ | $0.2536 \pm 0.0015$ | $\mathbf{0.3444 \pm 0.0017}$ |
| **S4G+** | $\mathbf{0.4036 \pm 0.0024}$ | $\mathbf{0.3421 \pm 0.0016}$ | $\mathbf{0.7293 \pm 0.0004}$ | $\mathbf{0.2461 \pm 0.0003}$ | $\mathbf{0.3374 \pm 0.0003}$ |

## 4.1 SYNTHETIC DATASET FOR OVER-SQUASHING

Tree-Neighbors-Match is a series of synthetic datasets from Alon & Yahav (2021), they use these datasets to simulate extreme cases of bottleneck on graphs, specifically learning features of the root nodes on *binary tree-shaped graphs* with different depths. For dataset of depth $r$, each sample is a binary tree with depth $r$, and each node contains two-dimensional features. For leaf nodes, they are their unique labels among all leaf nodes and the shuffled labels; for root nodes, the feature in the first dimension is the label of a certain leaf node in the graph, and the second dimension is filled with zeros waiting for model prediction; all other nodes are filled with zero-valued features. In order to complete this task, GNN needs to aggregate information from all the leaf nodes. This benchmark only focus on the training accuracy to check if a GNN model could fit the hard training cases. We test on $r$ from 2 to 8. We select the classical MPNNs as baselines: GGNN (Li et al., 2015), GCN (Kipf & Welling, 2017), GAT (Veličković et al., 2018), and GIN (Xu et al., 2019).

**Results and discussion.** As we can see in Table 1, all the baselines lose the ability to fit the long-range signals since $r = 4$ and perform badly when $r$ is large, indicating they suffer from severe bottleneck on graphs. However, our proposed S4G successfully fit the dataset almost perfectly on all depths. It is worth noting that although GAT and GGNN have filtering mechanisms for aggregating messages, which can alleviate the redundant information to some extent, they are still fundamentally limited by the sensitivity to exponential decay in MPNN frameworks. Our model has linear sensitivity, so it allows the information from leaf nodes to not decay too much, effectively handling LRI.

## 4.2 REAL-WORLD LONG RANGE TASKS

After showing the strong fitting ability for long-range signals on synthetic datasets, a natural question is how our model performs on real-world datasets requires LRI. Long Range Graph Benchmark (LRGB) (Dwivedi et al., 2022b) is a collection of 5 real-world datasets that emphasize the importance of long-range dependencies. LRGB covers major graph tasks such as graph regression, graph classification, link prediction, and node classification. It collects data from computer vision, chemistry, and quantum chemistry domains. For the classic methods used for comparison, we select GCN (Kipf & Welling, 2017), GINE (Xu et al., 2019), and GatedGCN (Bresson & Laurent, 2017) for classical MPNNs, and Transformer+LapPE for GT. We also compared to multi-hop MPNN and graph rewiring methods, since they are developed for LRI. MixHop (Abu-El-Haija et al., 2019) and SPN (Abboud et al., 2022) allows nodes to directly access higher-order neighbors in one message-passing step, and we choose it as a representative of multi-hop MPNN. DIGL (Klicpera et al., 2019) uses the static graph rewiring method by modifying the graph structure before running MPNN, and we use it to reflect the performance of graph rewiring methods. For state-of-the-art models, we select SAN (Kreuzer et al., 2021), GraphGPS (Rampášek et al., 2022), Exphormer (Shirzad et al., 2023), and

Table 3: Comparison of S4G, S4G+ with baselines on Benchmarking GNNs (Dwivedi et al., 2020). Best results are colored in **first**, **second** and **third**.

| Model | PATTERN | CLUSTER | MNIST | CIFAR10 |
|---|---|---|---|---|
| GCN (Kipf & Welling, 2017) | 71.89 ± 0.334 | 68.50 ± 0.976 | 90.71 ± 0.218 | 55.71 ± 0.381 |
| GAT (Veličković et al., 2018) | 78.27 ± 0.186 | 70.59 ± 0.447 | 95.54 ± 0.205 | 64.22 ± 0.455 |
| GIN (Xu et al., 2019) | 85.39 ± 0.136 | 64.72 ± 1.553 | 96.49 ± 0.252 | 55.26 ± 1.527 |
| GatedGCN (Bresson & Laurent, 2017) | 85.57 ± 0.088 | 73.84 ± 0.326 | 97.34 ± 0.143 | 67.31 ± 0.311 |
| GT | 50.78 ± 0.090 | 20.92 ± 0.038 | 97.33 ± 0.715 | 69.84 ± 0.918 |
| SPN | 86.57 ± 0.137 | 16.69 ± 0.075 | 83.31 ± 4.455 | 37.22 ± 8.265 |
| **S4G** | **86.87 ± 0.024** | **78.37 ± 0.085** | 96.37 ± 0.165 | 70.65 ± 0.328 |
| SAN (Kreuzer et al., 2021) | 86.58 ± 0.037 | 76.69 ± 0.65 | - | - |
| GraphGPS (Rampášek et al., 2022) | 86.69 ± 0.059 | 78.02 ± 0.180 | **98.05 ± 0.126** | **72.30 ± 0.356** |
| Exphormer (Shirzad et al., 2023) | **86.74 ± 0.015** | **78.07 ± 0.037** | **98.55 ± 0.039** | **74.69 ± 0.125** |
| **S4G+** | **86.88 ± 0.029** | **78.35 ± 0.199** | **97.65 ± 0.131** | **72.55 ± 0.379** |

DRew (Gutteridge et al., 2023). Among them, SAN simultaneously runs local attention on the original graph and global attention on a fully connected graph; GraphGPS integrates MPNN, GT, and PE/SE; Exphormer inherits the framework of GraphGPS but replaces its GT module with a sparse attention layer driven by an expander graph and virtual global nodes. DRew combines multi-hop MPNN with graph rewiring, resulting in the 'dynamic rewiring' models. Our S4G and S4G+ models share the same parameter budget as all baseline comparisons - not exceeding 500k parameters.

**Results and discussion.**   According to Table 2, our S4G model has shown superior performance compared to classical methods. Compared to classical MPNN and GT models, S4G achieved a significant advantage, leading by more than 7 points on the Peptides-func dataset. It also outperformed multi-hop MPNN and graph rewiring methods with an improvement of 3 points. Compared to the SOTA methods, our S4G+ still achieves leading performance, surpassing existing SOTA methods on three datasets. It is worth noting that although some SOTA models can match our performance on certain datasets, they exhibit significant gaps on other datasets. For example, Exphormer performs equally well as us on two CV datasets - PascalVOC-SP and COCO-SP, but there is a 7-point gap between us on the Peptides-func dataset. On the other hand, while DRew performs well on Peptides-func, it falls nearly 7 points below us on PascalVOC-SP. The consistent superior performance across five datasets indicates that our model has a clear advantage in handling long-range dependencies compared to previous methods.

### 4.3 GENERAL GNN BENCHMARKING

In previous experiments, we confirmed the model's strong ability to handle LRI using synthetic and realistic datasets. However, our proposed SSM convolution on graphs, a new paradigm distinct from message passing and attention mechanisms, has not been validated for modeling beyond LRI. We select 4 datasets from Benchmarking GNNs (Dwivedi et al., 2020) to evaluate the general ability for graph learning of our model. PATTERN and CLUSTER are two synthetic node classification datasets sampled from Stochastic Block Model, a popular model to generate communities. CIFAR10 and MNIST are two graph classification datasets constructed from computer vision area. We select GCN (Kipf & Welling, 2017), GAT (Veličković et al., 2018), GIN (Xu et al., 2019), GatedGCN (Bresson & Laurent, 2017), and Transformer+LapPE as the classic methods for comparison; SAN (Kreuzer et al., 2021), GraphGPS (Rampášek et al., 2022) and Exphormer (Shirzad et al., 2023) were chosen as the SOTA methods. These SOTA models have been introduced earlier. We also compared with SPN (Abboud et al., 2022), a recent work also aggregate based shortest path.

**Results and discussion.**   According to Table 3, we can see that S4G demonstrates significant advantages over the classical models in two node classification datasets, even approaching the performance of SOTA models. It provides a gain of more than 4 points compared to classical methods on the CLUSTER dataset. Additionally, it matches the performance of GT on graph classification datasets CIFAR10 and MNIST. Compared to the SOTA model Exphormer, S4G+ has also demon-

Table 4: Ablation study for S4G on PascalVOC-SP, Peptides-func for long-range tasks, and PATTERN, CIFAR10 for general graph tasks.

| Model | PascalVOC-SP | Peptides-func | PATTERN | CIFAR10 |
|---|---|---|---|---|
| **S4G** | **0.3183 ± 0.0078** | **0.7213 ± 0.0009** | **0.8687 ± 0.0002** | **0.7065 ± 0.0033** |
| random kernel | 0.1533 ± 0.0750 | 0.3880 ± 0.0153 | 0.4998 ± 0.0018 | 0.3358 ± 0.0036 |
| shuffle kernel | 0.2709 ± 0.0132 | 0.5780 ± 0.0051 | 0.8670 ± 0.0008 | 0.5456 ± 0.0060 |
| simple kernel | 0.2763 ± 0.0061 | 0.5814 ± 0.0187 | 0.8540 ± 0.0032 | 0.5846 ± 0.0045 |

strated competitive performance. These results indicate that our proposed convolution based on SSM as a new paradigm for learning on graphs provide performance improvement against message passing and attention mechanisms, showing great potential in general graph tasks.

### 4.4 ABLATION STUDY

In this section, we conduct an ablation study on the core component of S4G, S4G-Conv. We aim to address some questions regarding S4G-Conv as listed below and validate them with the experimental results presented in Table 4. Due to the key role of SSM kernel $\overline{K}$ in S4G-Conv, our ablation study will focus on it. We select 2 datasets from LRGB and 2 datasets from Benchmarking GNNs for testing the performance change in both long-range problem and general tasks.

- Is the performance improvement of the model really achieved by introducing SSM, rather than directly accessing neighbors?
- Is it really necessary to introduce SSM through modeling rooted-trees? Does the rooted-tree hierarchy truly encode structural information?
- Is it really necessary to introduce SSM to achieve linear sensitivity? Is there a simpler way that could work?

To answer the first question, we replaced the SSM kernel with noise sampled from a Gaussian distribution, and we refer to this variant as "random kernel". For the second question, we shuffled the values of each position in the SSM kernel to disrupt the hierarchical information encoded by the rooted tree, and obtain the variant "shuffle kernel". As for the last question, we designed a simplified alternative approach where $\overline{K}_{[d_{\mathcal{G}}(i,j)]}$ is directly replaced with $\frac{1}{1+d_{\mathcal{G}}(i,j)}$, which still ensures that gradients between nodes linearly decrease with distance, we called it "simple kernel".

**Results and discussion.** Based on Table 4 results, the random kernel has the lowest performance. The shuffle and simple kernels are similar, but both are lower than the original S4G. This shows that using random values alone is not effective for model learning. Encoding rooted-tree hierarchy allows for effective reflection of structural information on graphs, as seen by the significant decline in performance with shuffled SSM kernels. Additionally, employing a simple linear weight descent for convolution does not yield satisfactory results because it does not capture the structural information in graphs. These findings suggest that modeling a rooted-tree with S4 is both effective and necessary for achieving linearly decayed sensitivity and good overall performance.

## 5 CONCLUSION

We focus on the long-range interaction on graphs. To address the issue of exponentially sensitivity decay with distance, we developed a model based on structured state spaces. The theoretical guarantee ensures its linear sensitivity, allowing us to break bottleneck on graphs while preserving the inductive biases. Our approach achieves leading performance on both synthetic and realistic datasets for LRI. Moreover, in general graph tasks, it demonstrates improved performance compared to classical methods as a novel mechanism distinct from message passing and attention mechanism. This highlights its potential to compete with state-of-the-art models across various tasks. We discuss the scalability, expressivity and future works for our model in Appendix B due to the space limit.

## 6 ETHICS STATEMENT

In this work, we propose a novel graph learning model. Our model is a general graph learning model that does not make explicit assumptions about data distribution, thus avoiding the occurrence of (social) bias. However, our data-driven mechanism needs to learn from samples with labels, and it is possible to be misled by targeting corrupted graph structures or features, which can have a negative social impact because both internet data and social networks can be represented as graphs. We can develop corresponding ethical guidelines and constraints for the usage of our model to avoid such problems.

## 7 REPRODUCIBILITY STATEMENT

We provide an introduction to our used datasets and model configuration in the experimental section of our paper. The dataset is mentioned first in this section. In our Appendix, we report the hardware and software environment used in our experiments, the methods used for hyper-parameter tuning, and the steps for downloading, loading and pre-processing the datasets. In our released source code, we list the steps required to reproduce the results and provide yaml files that include all the hyperparameters for the reported performance in this paper.

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

## A  PROOF

According to Equation 9, the Jacobian of node $i$ with respect to node $j$ is the value of the SSM kernel, i.e., $\overline{K}_{[d_{\mathcal{G}}(i,j)]}$. According to Lemma 1, the HiPPO framework ensures that the gradient norm between two tokens at a distance $\ell$ is $\Theta(1/\ell)$. In the SSM kernel, we only introduce an additional parameter vector $C$. Assuming that the Jacobian of parameter vector $C$ is bounded and independent of the shortest path distance $r$, and the ADD operator do not decay the gradients when propagated through it, we have the sensitivity between $i, j$ to be $\|\partial h_i / \partial h_j\| = \Theta(1/r)$.

## B  DISCUSSION

**Scalability.** Although our model has theoretically linear complexity as stated in Section 3.3, in this paper, all the datasets we tested have relatively small graph sizes, with the largest graph not exceeding 512 nodes. Therefore, we did not impose any restrictions on the node budget of the model in practical experiments. In the future, it would be interesting to naturally extend S4G to large-scale graphs.

**Expressivity.** Our model is able to distinguish any pair of graphs that can be distinguished by the shortest path kernel, because it directly injects the shortest path information when constructing hop-wise sequences. For example, in the case of a hexagonal cycle and two triangles mentioned in Zhang & Li (2021), the classical MPNN cannot distinguish between these two graphs because their expressive power is limited by 1-WL. However, our model could distinguish them. We will consider further research on more refined expressive power as a future direction.

## C  EXPERIMENTAL DETAILS

Table 5: Dataset statistics

| Dataset | Graphs | Avg. nodes | Avg. edges | Prediction Level | No. Classes | Metric |
|---|---|---|---|---|---|---|
| PascalVOC-SP | 11,355 | 479.4 | 2,710.5 | inductive node | 21 | F1 |
| COCO-SP | 123,286 | 476.9 | 2,693.7 | inductive node | 81 | F1 |
| PCQM-Contact | 529,434 | 30.1 | 61.0 | inductive link | (link ranking) | MRR |
| Peptides-func | 15,535 | 150.9 | 307.3 | graph | 10 | Average Precision |
| Peptides-struct | 15,535 | 150.9 | 307.3 | graph | 11 (regression) | Mean Absolute Error |
| MNIST | 70,000 | 70.6 | 564.5 | graph | 10 | Accuracy |
| CIFAR10 | 60,000 | 117.6 | 941.1 | graph | 10 | Accuracy |
| PATTERN | 14,000 | 118.9 | 3,039.3 | inductive node | 2 | Accuracy |
| CLUSTER | 12,000 | 117.2 | 2,150.9 | inductive node | 6 | Accuracy |

In this section, we will discuss the experimental environment, dataset, hyperparameter tuning, and other experimental details.

### C.1  HARDWARE & SOFTWARE

We use NVIDIA GeForce RTX 3090 and NVIDIA GeForce RTX 4090 as the hardware environment, and use PyTorch and PyTorch Geometric (Fey & Lenssen, 2019) as our software environment. The datasets are downloaded from PyTorch Geometric, we also use dataloader provided by PyTorch Geometric to load and pre-process datasets. We fix the random seed for reproducibility. We report detailed configurations as follows.

### C.2  DATASETS & TUNING

For synthetic datasets, we follow the settings proposed by Alon & Yahav (2021), created a separate dataset for every tree depth (which is equal to $r$, the problem radius) and sampled up to 32,000 examples per dataset. The results of the baseline model were obtained directly from Alon & Yahav (2021). As for the S4G model, we only tune the learning rate without tuning other hyperparameters.

The statistical properties of the real-world dataset we are discussing in this article are presented in Table 5 and Table 2. We only tune the learning rate, dropout, weight decay, hidden size, and num layers. All the best hyperparameter for each dataset is stored as a yaml file and publish with our source code.

