# OpenReview forum: "S4G: Breaking the Bottleneck on Graphs with Structured State Spaces"
_ICLR.cc/2024/Conference — Submitted to ICLR 2024_

### Official Review · Reviewer_qfiX · 2023-10-23

**Soundness:** 3 good
**Presentation:** 3 good
**Contribution:** 3 good
**Rating:** 8
**Confidence:** 3

**Summary:**

In this paper, the authors propose a new graph NN architecture, S4G, using structured state space. S4G maintains similar inductive biases induced by regular MPNN but has the sensitivity with a linear decay. In this way, the message bottleneck problem is largely alleviated. Unlike transformer-based architecture, S4G does not need positional encodings or positional structures, which usually require a heuristic design. Empirical study shows that S4G consistently has superior performance over long-range tasks, which also corroborates their theoretical claims.

**Strengths:**

1. The paper is well-written for most of the part.
2. The proposed method appears novel and provides an effective way to solve the message bottleneck problem while maintaining the inductive bias induced by regular MPNNs.

**Weaknesses:**

The calculation of  $\bar{K}$ could be expensive and requires preprocessing for efficient training. A discussion on how much time is needed to preprocess for different datasets could make the results stronger.

**Questions:**

Please see the Weaknesses section.

---

> ### Author Response · Authors · 2023-11-17
> **Response to Reviewer qfiX**
>
> We are grateful for the valuable feedback and encouraging comments received regarding our paper. Below, we offer detailed responses to address your inquiries.
>
> > (Weakness) The calculation of $\bar{K}$ could be expensive and requires preprocessing for efficient training. A discussion on how much time is needed to preprocess for different datasets could make the results stronger.
>
> Following your suggestion, we report the pre-processing time for all datasets as follows.
>
> | **Time cost**     | **PATTERN** | **CLUSTER** | **MNIST** | **CIFAR10** | **PascalVOC-SP** | **COCO-SP** | **Peptides-func** | **Peptides-struct** | **PCQM-Contact** |
> | ----------------- | ----------- | ----------- | --------- | ----------- | ---------------- | ----------- | ----------------- | ------------------- | ---------------- |
> | **Per graph**     | 4.1ms       | 4.1ms       | 2ms       | 4.2ms       | 46ms             | 58ms        | 25ms              | 27ms                | 1.5ms            |
> | **Whole dataset** | 1min        | 0.8min      | 2.4min    | 4.2min      | 8.8min           | 118.4min    | 6.4min            | 7min                | 13.3min          |

---

> ### Author Response · Authors · 2023-11-20
> **The end of the discussion phase approaching**
>
> Dear Reviewer qfiX, as the discussion period comes to a close, we would like to thank you once again for your positive assessment. Your support and inspirational comments have been invaluable to us. We remain open and eager to incorporate any further feedback or insights you might offer.

---

### Official Review · Reviewer_rkT5 · 2023-10-31

**Soundness:** 2 fair
**Presentation:** 2 fair
**Contribution:** 1 poor
**Rating:** 3
**Confidence:** 4

**Summary:**

The paper proposes a new model based on structured state spaces, termed S4G, for enabling better information flow in graph neural networks without losing the graph inductive bias. The fundamental idea builds on two observations: (1) Graph neural networks have strong relational inductive bias, but they are subject to an exponential decay in information with increasing number of layers and (2) Graph transformers are subject to only constant decay in information but they typically lack appropriate graph inductive biases. The idea is to use "structured state spaces" to capture the hierarchical structure of rooted-trees from the source nodes, and to keep strong inductive bias while having only linear decay of information. Authors present experimental results on long-range and general graph benchmarks.

**Strengths:**

- **Problem setup**: It remains challenging to capture long-range interactions using graph neural networks for various reasons discussed in the paper, so the problem formulation is important and meaningful.

**Weaknesses:**

- **Scholarship**: The paper fails to present a good coverage of the related work which also makes the contributions questionable. This is particularly the case with the coverage of recent multi-hop approaches. Authors mention that multi-hop models do not help with e.g. over-squashing as they rely on taking the powers of the adjacency matrix which amplifies the over-squashing problem. This is true, but this is exactly the reason why other multi-hop approaches have been studied extensively, see e.g. [1], where the idea is to directly aggregate information from higher-order neighbours obtained using shortest path distances (a sensitivity analysis is also conducted).

- **Novelty, Originality, and Significance**: To the best of my understanding, the proposed idea of this paper appears to be largely covered by [1], since rooted trees are essentially constructed in the exact same way and aggregation is over the respective neighbourhoods $N_1...N_k$ of a particular node. Moreover, the graphormer model [2] follows essentially a very similar path: it aggregates over the neighbors at different shortest path distances directly. I do not see a fundamentally new or novel aspect in the present work, and neither a significant contribution. I'm happy to re-evaluate if the authors could better frame their approach in the existing literature and can identify the differences and contributions.

- **Experiments on Long Range Graph Benchmarks**: The empirical results do appear promising, but unfortunately, the benchmark of Dwiwedi et al has been criticised recently [3] and it turns out that the gap between GNNs and graph transformers either disappears or becomes insignificant after a systematic tuning of the GNN models. This is a very recent finding and the current paper cannot be held responsible, but given that this is one of the two experiments conducted, the validity of the proposal remains questionable. It is also unclear whether the above-mentioned approaches, i.e., graphormers, would match the presented results.

- **Technical limitations**: There are many limitations of the sensitivity analysis of Topping et al (and other approaches are proposed recently see eg  [4]). It requires bounded derivatives (which may not hold in practice) and also a normalised adjacency matrix. It is easy to see that without the assumption on the latter the values will explode rather than vanishing. On the other hand, it is easy to show that simple tricks (such as a fully connected layer, or adding a virtual node) can theoretically "maximize" this bound, and a systematic evaluation is needed against these simple model variations to clearly identify the benefit of the proposed idea.

[1] Aboud et al. Shortest Path Networks for Graph Property Prediction. LoG 2022.

[2] Ying et al., Do Transformers Really Perform Badly for Graph Representation? NeurIPS 2021.

[3] Tönshoff et al,  Where Did the Gap Go? Reassessing the Long-Range Graph Benchmark. 2023.

[4] Di Giovanni et al, How does over-squashing affect the power of GNNs? 2023.

**Questions:**

Please refer to my review.

---

> ### Author Response · Authors · 2023-11-17
> **Response to Reviewer rkT5 (Part 1/3)**
>
> We appreciate your thorough feedback and insightful inquiries. After careful consideration of your critique, we are committed to addressing your concerns and presenting arguments that justify a higher score for the paper.
>
> Before responding to your questions one by one, we would like to express our gratitude for mentioning SPN [1]. This work has a solid theoretical foundation and empirical analysis, and it addresses the issues encountered by multi-hop MPNN to a great extent, specifically the dependency on the power of the adjacency matrix, using a simple yet effective method. In our updated submission (please refer to the red-colored text), we introduce the works [1,2] you mentioned in our Introduction (paragraph 2 in Section 1). Furthermore, in the experimental part, we introduce SPN as a baseline (paragraph 1 in Section 4.2 and Section 4.3). We conducted experiments on both long-range and general datasets for SPN, and the results are presented in Table 2 and Table 3.
>
> > (Weakness 1) Scholarship: The paper fails to present a good coverage of the related work which also makes the contributions questionable. This is particularly the case with the coverage of recent multi-hop approaches. Authors mention that multi-hop models do not help with e.g. over-squashing as they rely on taking the powers of the adjacency matrix which amplifies the over-squashing problem. This is true, but this is exactly the reason why other multi-hop approaches have been studied extensively, see e.g. [1], where the idea is to directly aggregate information from higher-order neighbours obtained using shortest path distances (a sensitivity analysis is also conducted).
>
> Thank you for mentioning SPN, which will enrich our related works. In our updated submission (please refer to the red-colored text), we introduced and cited SPN in the Introduction (paragraph 2 in Section 1). In the experimental section, we used SPN as a baseline and conducted experiments on long-range and general datasets (paragraph 1 in Section 4.2 and Section 4.3). The experimental results are presented in Table 2 and Table 3.

---

> > ### Author Response · Authors · 2023-11-17
> > **Response to Reviewer rkT5 (Part 2/3)**
> >
> > > (Weakness 2) Novelty, Originality, and Significance: To the best of my understanding, the proposed idea of this paper appears to be largely covered by [1], since rooted trees are essentially constructed in the exact same way and aggregation is over the respective neighbourhoods $N_1...N_k$ of a particular node. Moreover, the graphormer model [2] follows essentially a very similar path: it aggregates over the neighbors at different shortest path distances directly. I do not see a fundamentally new or novel aspect in the present work, and neither a significant contribution. I'm happy to re-evaluate if the authors could better frame their approach in the existing literature and can identify the differences and contributions.
> >
> > [1,2] indeed aggregate messages based on the shortest path, however, our work differs significantly from them in terms of motivation, theoretical analysis, and solution. Since [2] can be considered as a special case of SPN [1], we mainly discuss the differences between our work and SPN.
> >
> > - Motivation: The focus of SPN is to enable direct communication between nodes and higher-order neighbors, without relying on the power of adjacency matrix. Our motivation is to design a model with linearly decayed sensitivity, thus striking a balance between MPNN (exponentially decayed) and GT (constant), while retaining the advantages of both.
> >
> > - Theoretical analysis: SPN also conducted sensitivity analysis. However, SPN did not achieve linearly decayed sensitivity, and they did not analyze the sensitivity of GT. Our work analyzes the sensitivity of both MPNN and GT, revealing that they are at two extremes, which motivate us to design a model with linearly decayed sensitivity. The method to achieve this (S4 [10]) is also theoretically guaranteed and principled.
> >
> > - Solution: SPN utilizes learnable weights to combine messages from different hops. In contrast, we convert the rooted tree into a sequence and employ a sequential model (S4) to aggregate messages. The key distinction lies in how we handle the information from each hop in node $i$'s perspective. If we denote the information in the $k$-th hop as $h_i^k$, SPN aggregates them in an **unordered** manner, treating $\{ h_i^{L}, h_i^{L-1}, \cdots, h_i^1 \}$ as a **set**. On the other hand, S4G treats them as a **sequence**: $(h_i^{L}, h_i^{L-1}, \cdots, h_i^1)$, and encodes them in an **ordered** manner using a sequential model. Consequently, S4G is able to capture more comprehensive structural information.
> >
> > - Contribution: To the best of our knowledge, S4G is the first work to extend S4 and SSM convolution to the graph domain. S4 has made significant breakthroughs in long-range modeling for sequence [10], image [11], point-cloud [12] and reinforce learning [13]. So it is valuable to introduce it into the graph domain to address over-squashing and long-range problems. Furthermore, our model does not rely on message passing and attention mechanisms, providing a new paradigm for GNN design.
> >
> > > (Weakness 3) Experiments on Long Range Graph Benchmarks: The empirical results do appear promising, but unfortunately, the benchmark of Dwiwedi et al has been criticised recently [3] and it turns out that the gap between GNNs and graph transformers either disappears or becomes insignificant after a systematic tuning of the GNN models. This is a very recent finding and the current paper cannot be held responsible, but given that this is one of the two experiments conducted, the validity of the proposal remains questionable. It is also unclear whether the above-mentioned approaches, i.e., graphormers, would match the presented results.
> >
> > - The work you mentioned is very new (it appeared on arXiv on 2023.9.1, while the ICLR deadline is 2023.9.28), so we have not fully considered the arguments of this work. We have checked this work and found that they have modified the features and performance metrics for some datasets, which we believe require further evaluation. We will thoroughly consider this work in the future. We would like to kindly emphasize that this work is exclusively available on arXiv after 5.28. Based on the ICLR 2024 Reviewer Guide, having a comprehensive discussion about this paper might go beyond the scope of our work.
> >
> > - Long Range Graph Benchmark [3] is widely used for studying long-range problems on graphs, and we achieved SOTA performance on it (Table 2). Additionally, we used synthetic datasets to demonstrate the effectiveness of our model in modeling LRI (Table 1). For the CLUSTER dataset in the general benchmarks (Table 3), we observed that SOTA models use very deep layers, such as SAN and GraphGPS with 16 layers, and Exphormer with 20 layers. This implies that long-range signals are also important in the CLUSTER dataset, and we achieve good results on it. Thus, it's convincing that S4G can handle LRI well. For the performance of [1,2], please refer to the results of SPN reported in our updated submission (Table 2 and Table 3).

---

> > > ### Author Response · Authors · 2023-11-17
> > > **Response to Reviewer rkT5 (Part 3/3)**
> > >
> > > > (Weakness 4) Technical limitations: There are many limitations of the sensitivity analysis of Topping et al (and other approaches are proposed recently see eg [4]). It requires bounded derivatives (which may not hold in practice) and also a normalised adjacency matrix. It is easy to see that without the assumption on the latter the values will explode rather than vanishing. On the other hand, it is easy to show that simple tricks (such as a fully connected layer, or adding a virtual node) can theoretically "maximize" this bound, and a systematic evaluation is needed against these simple model variations to clearly identify the benefit of the proposed idea.
> > >
> > > - Thank you for pointing out the limitations of [4]. We also acknowledge these limitations, despite the fact that this work has been referenced by many works focusing on the over-squashing problem ([3,5,6,7,8] and SPN [1]). Regarding the new approach [9] you mentioned, we've added it as a footnote on page 3 of our updated submission (please refer to the red-colored text).
> > >
> > > - However, the purpose of our paper is not to maximize the bound and prevent the decaying, but to provide linearly decaying sensitivity in a principled way, thus achieving a balance between long-range modeling capability and graph inductive bias. Therefore, even if the analysis in [4] does not hold for all MPNNs, it does not affect the main contribution of our proposed method.
> > > - As for the simple tricks like a fully connected graph or a virtual node you mentioned, they can indeed maximize this bound, but it may come at the cost of losing inductive bias on the graph and sacrificing performance. It is worth noting that the GT contains a fully connected attention graph, and Exphormer relies on the virtual node. The results of GT and Exphormer are already reported in Table 2 and Table 3. Our model achieves superior performance compared to them.
> > >
> > > ---
> > >
> > > [1] Shortest Path Networks for Graph Property Prediction, LoG 2022
> > >
> > > [2] Do Transformers Really Perform Badly for Graph Representation? NeurIPS 2021
> > >
> > > [3] Long Range Graph Benchmark, NeurIPS 2022
> > >
> > > [4] Understanding over-squashing and bottlenecks on graphs via curvature, ICLR 2022
> > >
> > > [5] Recipe for a general, powerful, scalable graph transformer, NeurIPS 2022
> > >
> > > [6] On Over-Squashing in Message Passing Neural Networks: The Impact of Width, Depth, and Topology, ICML 2023
> > >
> > > [7] Drew: Dynamically rewired message passing with delay, ICML 2023
> > >
> > > [8] Exphormer: Sparse transformers for graphs, ICML 2023
> > >
> > > [9] How does over-squashing affect the power of GNNs? arXiv 2023
> > >
> > > [10] Efficiently Modeling Long Sequences with Structured State Spaces, ICLR 2022
> > >
> > > [11] S4ND: Modeling Images and Videos as Multidimensional Signals with State Spaces, NeurIPS 2022
> > >
> > > [12] Modelling Long Range Dependencies in D: From Task-Specific to a General Purpose CNN, ICLR 2023
> > >
> > > [13] Structured State Space Models for In-Context Reinforcement Learning, NeurIPS 2023

---

> ### Author Response · Authors · 2023-11-20
> **The end of the discussion phase approaching**
>
> Dear Reviewer rkT5, as the discussion period draws to a close, we would like to verify if our responses adequately address your inquiries. In light of what you've mentioned, we introduced [1,2] in our paper and conducted experiments for [1]. Additionally, we discussed the difference between our model and SPN [1] in detail. All the other questions received a direct response as well. We greatly appreciate your valuable feedback and recommendations to enhance our paper. We eagerly await your response.
>
> ---
>
> [1] Shortest Path Networks for Graph Property Prediction, LoG 2022
>
> [2] Do Transformers Really Perform Badly for Graph Representation? NeurIPS 2021

---

> > ### Comment · Reviewer_rkT5 · 2023-11-21
> > **Thanks for the additional work**
> >
> > I appreciate the additional work by the authors. I understand the differences from SPNs and similar models, but I am not convinced by the virtue of the proposed approach. There is a need for a much more systematic study to make this convincing in terms of the information flow, expressive power, and alike. I do not find the reported results for SPN very consistent or reproducible either. In fact, some results look fairly low which may be due to poor choice of hyper-parameters and alike. We have no access to the hyper-parameters and not even to the distance parameter used in these experiments. All in all, I do not see a good reason to change my initial assessment of the paper.

---

> > > ### Author Response · Authors · 2023-11-21
> > > **Further response to Reviewer rkT5**
> > >
> > > Dear reviewer rkT5, thank you for your time and feedback.
> > >
> > > - Regarding the issue of expressive power, our work does not focused on the expressivity of GNNs, and the purpose of the paper is not to design a highly expressive GNN model, despite S4G is more powerful than MPNN. For more details, please refer to *Response to Reviewer SDke - Weakness 2*.
> > >
> > > - Regarding the issue of SPN experimental results, the hyper-parameters for SPN are now included in our released code. SPN has not been experimented on any dataset we tested in this paper. Considering the time constraints for the rebuttal period and the similarity between SPN and S4G as you mentioned, we selected hyper-parameters similar to S4G, except for increasing the hidden dimension in LRGB to ensure that SPN fully utilizes the parameter budget. As for the distance parameter for SPN, we allow SPN to capture the full graph information, which is the same as S4G.
> > >
> > > - In the limited time for rebuttal, we did not have sufficient resources to perform thorough hyper-parameter tuning for the new baseline you mentioned, i.e., SPN, on each dataset. If the paper is accepted, we will conduct more thorough tuning for SPN and report its optimal performance in the camera-ready version. Considering SPN was published in 2022, and we have already compared S4G to several SOTA methods published in 2023, we believe the experimental results are convincing enough.

---

### Official Review · Reviewer_SDke · 2023-11-01

**Soundness:** 2 fair
**Presentation:** 3 good
**Contribution:** 2 fair
**Rating:** 3
**Confidence:** 4

**Summary:**

This paper proposes a new architecture for graph representation learning to address the limitations of message passing neural networks. Specifically, to generate the representation for a target node, representations of nodes that are at the same hop are summed up, and then a structured state space model (S4) is applied to the sequence of hop representations.
This paper demonstrates the long-range modeling capacity of the proposed architecture by analyzing the sensitivity between distant nodes. The proposed model shows good empirical performance on a series of graph benchmarks.

**Strengths:**

- The idea of extending S4 to graphs is interesting and novel to the graph ML domain.
- The sensitivity analysis is intuitive and clear to explain why S4 can help to utilize the information of distant nodes.
- The model shows good empirical performance.

**Weaknesses:**

- Over-smoothing and over-squashing: The paper claims that the proposed model can address both over-smoothing and over-squashing. However, over-smoothing is caused by the lack of local neighborhood information. S4 aims to better capture distant information, which seems to be in the opposite direction of addressing over-smoothing. The experiments didn’t touch over-smoothing either.
- Lack of expressiveness analysis: By converting a neighborhood into a sequence, the model considers the shortest distance, but would inevitably lose other structural information. E.g., the model doesn’t know the edges between hop $k-1$ and hop $k$, i.e., for a certain node at hop $k-1$ which nodes at hop $k$ are connected to it. This questions the expressiveness of the proposed model. Theoretical analysis is necessary to justify the expressiveness and support the empirical results, but it’s missing.
- According to the experiments, the performance of S4G itself is not good enough on several datasets, while S4G+, which contains an extra MPNN layer, can do significantly better. This implies that S4G itself may lose some structural information (together with the above point)

**Questions:**

- More details of the experimental setup should be given, such as hyper-parameters (e.g., is the hidden dimension very large? (since the HIPPO matrix is fixed))
- What is the specific difference for those baselines with asterisk? Did the proposed model follow the asterisk setting?

---

> ### Author Response · Authors · 2023-11-17
> **Response to Reviewer SDke (Part 1/2)**
>
> Thank you for your detailed comments and valuable questions. We have carefully considered your criticism and aim to persuade you that the paper deserves a higher score by addressing your concerns.
>
> > (Weakness 1) Over-smoothing and over-squashing: The paper claims that the proposed model can address both over-smoothing and over-squashing. However, over-smoothing is caused by the lack of local neighborhood information. S4 aims to better capture distant information, which seems to be in the opposite direction of addressing over-smoothing. The experiments didn’t touch over-smoothing either.
>
> - Our paper mentions the over-smoothing problem, but the focus of this paper is on the over-squashing and long-range problems (as emphasized in our title, abstract, and conclusion). The over-smoothing problem has been largely addressed in previous works, so we did not design experiments for it.
>
> - S4G is not on the opposite side of over-smoothing: S4G encodes neighbors from low to high order into a sequence. Based on the memory capacity of the HiPPO framework [1], the model avoids forgetting local information, thus avoiding the over-smoothing problem. On the other hand, the main cause of the over-smoothing problem is that MPNN is similar to Laplacian smoothing [2], which leads to multiple layers of MPNN degenerating into a low-pass filter on the graph [3]. S4G does not perform message passing on the original graph structure, so it can avoid such degeneration.
>
> - It is worth noting that we have already used a 16-layer model on our tested CLUSTER dataset, a general node classification dataset. This indicates that the performance does not decrease with depth and avoids over-smoothing issues. To further alleviate your concerns, we tested the performance of S4G from 2 to 10 layers on another node classification dataset we tested, i.e., PATTERN. The results are as follows, the performance remains stable as the number of layers increases.
>
>     | **# Layers** | **2** | **4** | **6** | **8** | **10** |
>     | ------- | ------- | ------- | ------- | ------- | ------- |
>     | **S4G** | $86.72 \pm 0.03$ | $86.87 \pm 0.02$ | $86.83 \pm 0.04$ | $86.85 \pm 0.02$ | $86.85 \pm 0.03$ |
>
> > (Weakness 2) Lack of expressiveness analysis: By converting a neighborhood into a sequence, the model considers the shortest distance, but would inevitably lose other structural information. E.g., the model doesn’t know the edges between hop $k-1$ and hop $k$, i.e., for a certain node at hop $k-1$ which nodes at hop $k$ are connected to it. This questions the expressiveness of the proposed model. Theoretical analysis is necessary to justify the expressiveness and support the empirical results, but it’s missing.
>
> - Our paper did not focus on the expressivity problem, we did not aim to design a expressive GNN in this work. However, we mentioned it in the conclusion and already provided some analysis in Appendix B.
>
> - Our model has stronger expressive power than MPNN. The following is a brief proof, and a comprehensive exploration of the expressive power of S4G will be part of future work.
>     - The information captured by S4G is more abundant than MPNN: In MPNN, under each node's perspective, each message passing aggregating information from the first-order star-shaped graph [4]. By contrast, S4G not only aggregates information from first-order neighbors but also from higher-order neighbors. Therefore, S4G can obtain more abundant information.
>     - There exists a pair of graphs that MPNN cannot distinguish but S4G could (this example is borrowed from [5]): Consider two uncolored graphs, G1 and G2, where G1 is a hexagon and G2 consists of two disconnected triangles. With the shortest path information encoded in the SSM kernel, S4G can differentiate between G1 and G2, while the 1-WL test cannot. Since the 1-WL test is the upper bound of the expressive power of MPNN, MPNN also cannot distinguish them.

---

> > ### Author Response · Authors · 2023-11-17
> > **Response to Reviewer SDke (Part 2/2)**
> >
> > > (Weakness 3) According to the experiments, the performance of S4G itself is not good enough on several datasets, while S4G+, which contains an extra MPNN layer, can do significantly better. This implies that S4G itself may lose some structural information (together with the above point)
> >
> > - Compared to single models such as MPNN, multi-hop MPNN, graph Transformer, and graph rewiring, S4G has advantages in both long-range and general datasets. For example, compared to the best-performing single method on the Peptides-func dataset, it achieved a relative improvement of 5.4%.
> >
> > - Comparing S4G directly with SOTA models is unfair, because SOTA models are hybrid models that incorporate multiple methods (see Section 4.2 for their introduction). If we simply combine S4G with MPNN to create S4G+, we demonstrate superior performance compared to SOTA on the long-range dataset, and we also match the SOTA on the general dataset.
> >
> > - Regarding the significant improvement of S4G+ compared to S4G, mainly on the PascalVOC-SP and COCO-SP datasets, we believe this is related to the nature of the datasets. Both of them are constructed from semantic segmentation datasets in computer vision, which may rely on both local and global signals simultaneously. For example, while most segmented objects have local characteristics, background objects are scattered globally (see Figure 2 in [6]). The degree of reliance on the global signal in these two datasets is not yet clear, as mentioned in [7] (see Section 7). It is worth noting that GraphGPS, as a hybrid model of GT and GatedGCN, also shows significant performance improvement compared to GT and GatedGCN on these two datasets, which partially confirms our hypothesis.
> >
> > ---
> >
> > > (Question 1) More details of the experimental setup should be given, such as hyper-parameters (e.g., is the hidden dimension very large? (since the HIPPO matrix is fixed))
> >
> > We have listed the source code, hyper-parameters and experimental steps required to reproduce the results of this paper in the released repository. In all our experiments, the state dimension of HiPPO is set to 64, while the hidden dimension of the neural network varies from 32 to 192.
> >
> > > (Question 2) What is the specific difference for those baselines with asterisk? Did the proposed model follow the asterisk setting?
> >
> > As stated in Section 4, model with asterisks indicate that it has different structural variants for each datasets, which can be reflected in the choice of PE/SE, MPNN, and GT. For models with asterisks, we select the best variant for each dataset and combine the results into a row in the table. Our model does not adjust the model structure across all datasets. Specifically, S4G+ only uses GCN on the PATTERN and CLUSTER, while GatedGCN is used on all other datasets. This is because these two datasets do not contain edge features, so we choose the simpler GCN.
> >
> > ---
> >
> > [1] HiPPO: Recurrent Memory with Optimal Polynomial Projections, NeurIPS 2020
> >
> > [2] Deeper Insights into Graph Convolutional Networks for Semi-Supervised Learning, AAAI, 2018
> >
> > [3] Simplifying Graph Convolutional Networks, ICML 2019
> >
> > [4] From Stars to Subgraphs: Uplifting Any GNN with Local Structure Awareness, ICLR 2021
> >
> > [5] Nested graph neural networks, NeurIPS 2021
> >
> > [6] Long Range Graph Benchmark, NeurIPS 2022
> >
> > [7] DRew: Dynamically Rewired Message Passing with Delay, ICML 2023

---

> ### Author Response · Authors · 2023-11-20
> **The end of the discussion phase approaching**
>
> Dear Reviewer SDke, as the discussion period ends soon, we would like to check whether our responses answer your questions. Following your comments, we conducted experiments to test the model's performance when the number of layers increases. Additionally, we provided a brief proof for the expressivity of S4G. All the other questions received a direct response as well. Thank you again for your comments and suggestions to improve our paper, and we look forward to your reply.

---

> ### Comment · Reviewer_SDke · 2023-11-23
> **Comment by Reviewer**
>
> Thank you for further clarification! However, my major concern, i.e., theoretical analysis of expressiveness which is necessary for a sequential model over graph-structured data, is not addressed. Therefore, I still don't recommend acceptance.

---

> ### Author Response · Authors · 2023-11-23
> **Response to Reviewer SDke**
>
> Dear Reviewer SDke, thank you for your response. It is important to emphasize that our research does not specifically delve into the expressive power of GNNs. The investigation into GNN expressive power stands as an orthogonal research direction to our focus. Our primary focus centers around addressing challenges related to long-range interactions (LRI) and over-squashing on graphs. Discussing the expressive power of S4G at length would deviate from our research question. As evidence, existing works that focus on the over-squashing issue and LRI, such as [1,2,3,4], have also not discussed the issue of expressive power in depth, work [1,2] are theoretical works, while [3,4] design models. Nevertheless, our response already demonstrates that our model is more powerful compared to MPNNs.
>
> ---
>
> [1] Understanding over-squashing and bottlenecks on graphs via curvature, ICLR 2022
>
> [2] On Over-Squashing in Message Passing Neural Networks: The Impact of Width, Depth, and Topology, ICML 2023
>
> [3] Drew: Dynamically rewired message passing with delay, ICML 2023
>
> [4] Exphormer: Sparse transformers for graphs, ICML 2023

---

### Author Response · Authors · 2023-11-17
**Source code release**

You can view the source code, hyper-parameters and experimental steps required to reproduce the results of this paper at the following anonymous link: https://anonymous.4open.science/r/S4G_ICLR24_SourceCode-1315.

---

### Meta-Review · Area_Chair_uAnZ · 2023-12-10

**Metareview:**

The paper presents a novel GNN architecture aimed at addressing the inherent limitations of message-passing neural networks (MPNNs) in capturing long-range interactions. The paper identifies the "over-squashing" problem in traditional GNNs and proposes the integration of S4 to capture hierarchical structures of rooted trees, achieving linear sensitivity with theoretical guarantees. The authors argue that their approach retains the strong inductive biases of MPNNs while also incorporating the long-range modeling capabilities of graph Transformers (GTs).

Strengths:
- The idea of extending S4 to graphs as presented in S4G, is incrementally novel.
- The sensitivity study appears sound.
- The model shows promising empirical performance across a range of graph benchmarks.

Weaknesses:
- The authors claim to address both over-smoothing and over-squashing issues. However, the focus seems more on the latter, with limited discussion and experimental evidence on the over-smoothing problem.
- The paper does not provide an in-depth analysis of the expressiveness of the proposed model. Reviewers raised this as an important limitation of the approach.
- The sensitivity analysis, while comprehensive, may have limitations in practice, such as the requirement of bounded derivatives and a normalized adjacency matrix.
- Some reviewers raised concerns about the specifics of the experimental setup, including the choice of hyper-parameters and the reproducibility of the SPN results.

Based on the discussions between authors and reviewers, the AC votes for rejection and encourages the authors to take into account the feedback from the reviewers in their next submission.

**Justification For Why Not Higher Score:**

Reviewers' major concerns were not addressed during rebuttal.

**Justification For Why Not Lower Score:**

N/A

---

### Decision · Program_Chairs · 2024-01-16

Reject